# 3D Culture Systems for Exploring Cancer Immunology

**DOI:** 10.3390/cancers13010056

**Published:** 2020-12-28

**Authors:** Allison A. Fitzgerald, Eric Li, Louis M. Weiner

**Affiliations:** Department of Oncology, Georgetown Lombardi Comprehensive Cancer Center, Georgetown University Medical Center, Washington, DC 20057, USA; ao546@georgetown.edu (A.A.F.); el822@georgetown.edu (E.L.)

**Keywords:** organoids, spheroids, tumor immunology, three-dimensional culture, microfluidic chips, immunotherapy

## Abstract

**Simple Summary:**

To study any disease, researchers need convenient and relevant disease models. In cancer, the most commonly used models are two-dimensional (2D) culture models, which grow cells on hard, rigid, plastic surfaces, and mouse models. Cancer immunology is especially difficult to model because the immune system is exceedingly complex; it contains multiple types of cells, and each cell type has several subtypes and a spectrum of activation states. These many immune cell types interact with cancer cells and other components of the tumor, ultimately influencing disease outcomes. 2D culture methods fail to recapitulate these complex cellular interactions. Mouse models also suffer because the murine and human immune systems vary significantly. Three-dimensional (3D) culture systems therefore provide an alternative method to study cancer immunology and can fill the current gaps in available models. This review will describe common 3D culture models and how those models have been used to advance our understanding of cancer immunology.

**Abstract:**

Cancer immunotherapy has revolutionized cancer treatment, spurring extensive investigation into cancer immunology and how to exploit this biology for therapeutic benefit. Current methods to investigate cancer-immune cell interactions and develop novel drug therapies rely on either two-dimensional (2D) culture systems or murine models. However, three-dimensional (3D) culture systems provide a potentially superior alternative model to both 2D and murine approaches. As opposed to 2D models, 3D models are more physiologically relevant and better replicate tumor complexities. Compared to murine models, 3D models are cheaper, faster, and can study the human immune system. In this review, we discuss the most common 3D culture systems—spheroids, organoids, and microfluidic chips—and detail how these systems have advanced our understanding of cancer immunology.

## 1. Introduction

Cancer immunotherapy represents a scientific breakthrough. Treatments such as immune checkpoint inhibitors, chimeric antigen receptor (CAR) T cells, and cytokine therapy, among others, are extending patients’ lives and in some cases offering cures. While each treatment works through a different mechanism, all cancer immunotherapies have the same goal—to enhance the patient’s own immune system to recognize and eliminate the cancer. The FDA has approved immunotherapy for at least 19 different cancer types. In 2019 alone, the FDA approved 15 immunotherapy regiments [1].

Despite the remarkable boom in available cancer immunotherapies, there is still a wealth of ongoing research aimed at improving existing immunotherapies or identifying new ones. In order to successfully do either, researchers must broaden and deepen their understanding of cancer immunology. Most research investigating novel concepts in onco-immunology depends on models such as mouse models or two-dimensional (2D) cell culture, both of which have limitations.

2D cell culture has been the method of choice for studying cancer cell biology and drug discovery since 1951, when a scientist at Johns Hopkins University obtained a sample of cervical cancer cells from a Black woman named Henrietta Lacks, without her consent as 1951 predates the concept of informed consent [2]. These cells were termed “HeLa” cells and their ability to grow indefinitely transformed cancer research. Scientists can now culture many cell types including immortalized cancer cell lines, immune cells, even primary human cells [3]. 2D cell culture offers many benefits, including low-cost, high-throughput capability, and the ability to use human cells to study human disease. However, this technique still requires growing cells on hard, rigid, plastic surfaces—conditions far removed from the tumor microenvironment that sustains cancer cell growth in physiological conditions. Under normal tumor circumstances, the tumor microenvironment consists of a heterogeneous and complex mix of cell types and extracellular matrix. A growing number of studies demonstrate that 2D culture systems severely alter cellular phenotypes and physiology [4,5]. This could partially explain why only 16% of drugs developed based on results in 2D systems find success in phase II and phase III clinical trials, with cancer therapies representing a substantial proportion of the failures [4].

Murine models better recapitulate the physiologic conditions of tumor growth. Researchers can grow malignant tumors in mice in one of two ways: (1) malignant cells can be injected into the mice or (2) mice are genetically engineered to develop a malignant tumor over a specific course of time or in response to certain stimuli. Either way, the tumors that develop are surrounded by a tumor microenvironment that is absent in 2D cultures—a clear benefit. However, the murine tumor microenvironment does not fully replicate the human tumor microenvironment [5]. Moreover, these tumors are often derived from murine cancer cells. Human cancer cells can be used, but in these models the mice must lack competent immune system (termed immunodeficient) in order to prevent rejection of the human cancer cells by the murine immune system, thus precluding the possibility of studying human immune cell interactions with human cancer cells in the context of the microenvironment. The relatively recent development of humanized mice circumvents some aspects of this challenge. In humanized mice models, an immunodeficient mouse is engrafted with human immune and hematopoietic cells that reconstitute the immune system. These mice can sustain human T cells, B cells and dendritic cells and produce high levels of human IgG and IgM [6]. While these mice provide an advanced model to study human immune cell-tumor interactions, they are difficult to generate and costly, thereby limiting their use.

Three-dimensional (3D) models represent a third preclinical approach to study cancer immunology. 3D models offer more physiological environments as compared with 2D models. Unlike murine models, 3D models are more amenable to low cost and high-throughput research needs and can use human cancers and immune components. 3D models have been used to advance many fields of cancer immunology research (Appendix A). The most common 3D models include spheroids, organoids, and microfluidic chips (Figure 1).

By the strictest definition, organoids are mini-organ-like clusters grown from stem or progenitor cells. These progenitor cells expand and differentiate to make multicellular and heterogeneous clusters containing cell types with similar phenotypes to the original human organ from which the progenitor cells were derived (Figure 2). Of note, both spheroids and organoids can be hollow or solid [7,8,9,10]. Organoids are commonly grown by either embedding them in a matrix or by culturing them in air-liquid-interface systems, although other methods such as spinner bioreactors have been described (Figure 3) [11]. In cancer research the term “organoid” has been expanded to include tumor-like cell clusters that are grown from tumor specimens [12,13,14]. Some papers describe these tumor-derived multicellular clusters as “tumoroids” [15].

As opposed to organoids, spheroids are not derived from progenitor cells but rather created by aggregating already differentiated cells—most commonly cancer cell lines (Figure 2). Spheroids usually do not require patient derived tissue, making them highly accessible but less physiologically relevant compared to organoids. Spheroids are made using techniques that allow cells to aggregate with one another but prevent contact with culturing materials. Common techniques to create spheroids include low adhesion plates or hanging droplet methods (Figure 3). Some researchers create spheroids by embedding cell lines in 3D matrix, similar to organoids [16,17]. Thus, some techniques can be used to create both spheroids and organoids. The terms “spheroid” and “organoid” have occasional overlap; some researchers show cell line aggregates can create organoid-like structures [18]. Other researchers have referred to cell line aggregates as “tumoroids” [10,17,19]. For clarity, throughout this review we will refer to multicellular aggregates generated from cancer cell lines as spheroids and multicellular aggregates generated from progenitor cells or tumor specimens as organoids.

Biotechnology devices, such as microfluidic chips, are engineered to enhance current 3D culture methods by allowing greater control and complexity in the experimental conditions. These devices can mimic vasculature, allow laminar flow, create cellular partitions, and regulate matrix stiffness, among other advances [19,20,21].

3D culture methods have been used extensively to study cancer biology in multiple cancer types, ultimately revealing novel insights into cancer-related fields such as hypoxia and angiogenesis, invasion and metastases, drug screening, mutagenesis, and more [17,22,23,24,25,26,27]. In this review we will focus on how researchers have used 3D culture systems to advance our understanding of cancer immunology and immunotherapy.

## 2. Spheroids

### 2.1. Immunotherapy Penetrance into Solid Tumors

Researchers use high-throughput drug screens to test multiple compounds against multiple cancer types, usually using cancer cell lines, to identify new treatment. Spheroids are convenient models for high-throughput drug screening because they are low cost and technically simple yet better simulate tumor attributes like hypoxia, dormancy, and drug resistance than 2D cultures [28,29]. For instance, Senkowski et al. screened 1600 drugs for efficacy against glucose-deprived colorectal cancer spheroids to better model drug efficacy in low nutrient tumor environments [30]. Unfortunately, the cellular homogeneity of the spheroids limits their use for immunotherapy screening because most effective immunotherapies require immune cells. However, homogenous spheroids can be used to further immunotherapy development for a specific drug development problem: drug penetrance. The inability to attain high tumoral drug concentrations, frequently due to dose limiting peripheral toxicities, hampers most cancer therapies, including immunotherapy. Poor tumor infiltration especially plagues antibody therapy [31]. Unlike small molecule inhibitors that readily penetrate tumors, larger monoclonal antibodies can have heterogeneous tumor distribution which can ultimately limit their therapeutic efficacy. Engineering antibodies to have greater tumor penetrance is an active and ongoing field of research but the techniques used to assess antibody penetrance are limited [31,32,33]. It is impossible to use 2D culture systems to assess antibody penetration because the cells are arrayed in a single layer, yet in vivo studies are time consuming and expensive. Spheroids offer a promising alternative to cut the cost and time associated with in vivo screening by providing a quick and convenient method to assess antibody penetration into a tumor mass in vitro.

Matrix-assisted-laser-desorption-ionization mass-spectrometry imaging (MALDI-MSI) is a widely employed imaging technique that can visualize the spatial arrangement of drugs and metabolites in tissue from the whole-body level to the subcellular level [34]. In a proof-of-concept study, Liu et al. demonstrated that a MALDI-MSI technique can detect and assess the distribution of cetuximab, an anti-EGFR monoclonal antibody, throughout colon cancer spheroids [35]. Since the MALDI-MSI technique can also be applied to tissue, it is feasible to identify optimal antibodies via a spheroid screening assay, then assess the best antibody candidate in vivo.

Of course, homogeneous spheroids do not fully recapitulate the complex tumor microenvironment in vivo, so the physiological relevance and clinical translatability of these spheroid screening assays is questionable. To address this concern, Rodallec et al. compared the efficacy of a penetration-enhancing immunoliposome they had developed in an in vitro spheroid models and in an in vivo murine model [36]. Immunoliposomes are liposomes engineered to contain monoclonal antibodies or antibody fragments that target the liposome to antigen expressing cells [37]. These authors engineered an immunoliposome with the HER2-targeting monoclonal antibody, trastuzumab, embedded in a liposomal membrane that encapsulated the chemotherapy agent, docetaxel. This immunoliposome had similar antiproliferative effects on breast cancer cell line spheroids in vitro as it did on MDA-MB-231 breast cancer tumors in mice. In both models, the immunoliposome reduced tumor cell growth better than either trastuzumab or docetaxel alone. This study illustrates that 3D spheroid models of drug response, particularly penetration-dependent drugs, can replicate in vivo findings, suggesting that spheroids may provide a useful method for drug penetration optimization.

### 2.2. Immune Cell Migration and Tumor Infiltration

Immune cell trafficking to and infiltration into tumors is critical for immunotherapy efficacy in solid malignancies. However, the exact mechanisms that regulate immune cell infiltration into tumors remain unclear. 2D culture methods are poorly suited to investigate infiltration because of their monolayer design. For example, Mark et al. recently demonstrated that cryopreserved natural killer (NK) cells retain their cytotoxicity properties in 2D assays but have a 5.6 fold reduction in cytotoxicity in 3D assays because of impaired migration, which was undetectable in the 2D assay [38]. 3D culture methods can more accurately model immune cell movement through extracellular matrix and tumor masses. Using spheroids derived from colon cancer cell lines, Courau et al. found that IL-15 enhanced donor-derived NK and cytotoxic T cell infiltration into spheroids increasing tumor cell lysis [39]. Increased NK cell spheroid infiltration was accompanied by increased expression of NK cell inhibitory receptors. The addition of monoclonal antibodies that block these inhibitory receptors (anti-MICA/B and anti-NKG2A) further increased spheroid cell lysis. This builds on previous work demonstrating that anti-MICA/B antibodies reduce the number of melanoma metastasis in an NK cell dependent manner in murine models [40]. In contrast to these murine models, Courau et al.’s work demonstrates the feasibility of using anti-MICA/B antibodies with human immune and tumor cells. Additional murine studies are needed to determine if IL-15 can enhance immune cell infiltration into solid tumors and to determine if IL-15 would enhance anti-MICA/B or anti-NKG2A targeting antibodies.

### 2.3. Mechanisms of Immunoediting: Tumor Dormancy and Immunosuppressive Microenvironment

Cancer immunoediting is the process cancer cells employ to evade the immune system. Cancer immunoediting has three phases: elimination, equilibrium, and escape (Figure 4) [41]. If the host immune system fails to eliminate cancer cells, the cancer cells may enter equilibrium where they remain present without growth or elimination. Eventually cancer cells can escape the immune system to become clinically detectable disease. Cancer cells can attain equilibrium or escape through a variety of mechanisms including inducing cellular dormancy, immunosuppression, and deficient antigen presentation [42].

The homogenous nature of spheroids can be exploited to create highly controlled experimental conditions. To explore immune cell-cancer cell interactions in a spheroid system, researchers can add exogenous immune components, such as immune cells derived from cell lines or from healthy human donors, or even cytokines that mimic immune cell presence. The limited number of cellular components in these 3D cocultures grants greater clarity into specific cell-cell interactions that are nearly impossible to deconvolute in complex in vivo experiments. Further, cell-cell interactions in spheroids are more physiologic than cell-cell interactions in 2D culture systems. Therefore, spheroid co-cultures provide convenient and interpretable methods for exploring immune cell-cancer cell crosstalk. Spheroid immune cell-cancer cell cocultures have revealed novel and potentially exploitable mechanisms behind cancer immunoediting.

For example, interferon gamma (IFNγ) is an anti-tumor cytokine released by activated T and NK cells. IFNγ participates in cancer cell elimination by inhibiting cancer cell proliferation, enhancing apoptosis, abrogating tumor angiogenesis, and activating certain immune cells [43]. By exposing murine melanoma spheroids to IFNγ, Liu et al. identified how tumors avoid IFNγ-induced apoptosis and proposed a novel way to block melanoma tumors’ transitions from elimination to equilibrium [44]. Liu et al. found that in response to IFNγ, stem-like subpopulations in melanoma spheroids overexpress the metabolic proteins IDO1 and AhR, which ultimately suppress cell death and instead activate a tumor dormancy program. An IDO1/AhR inhibitor reversed the tumor dormancy response and resulted in reduced spheroid growth after exposure to IFNγ in vitro. The same treatment regimen of IDO1/AhR inhibitor with IFNγ reduced tumor growth and prolonged the overall survival of NOD/SCID mice bearing B16 melanoma tumors. Importantly, Liu et al. showed that the IFNγ resistance was only present when the B16 melanoma cells were cultured in 3D. When cultured in 2D, nearly all the B16 melanoma cells died in response to IFNγ—highlighting the importance of using 3D cultures when exploring cancer cell biology in vitro.

Transforming growth factor beta (TGF-β) is an immunosuppressive cytokine also involved in immunoediting. Similar to Liu et al., Stuber et al. used exogenous cytokine to interrogate its function. Unlike Liu et al., Stuber exposed the immune cells (in this study CAR T cells), not the cancer cells, to exogenous cytokine [45]. In vitro, TGF-β reduced the cytolytic activity, cytokine production and proliferation of CAR T cells, which translated to reduced lysis of breast cancer cells in 2D. A TGF-β inhibitor reversed the immunosuppressed phenotype of the CAR T cells and restored breast cancer cell lysis. These authors validated their findings in 3D spheroid models of their breast cancer line. While the results from the 3D experiments were not significantly different from the 2D experiments, the 3D experiments provided additional evidence that the use of a TGF-β inhibitor may potentiate CAR T cell efficacy in vivo and in the clinic.

Creating an immunosuppressive microenvironment is critical for cancer cells to escape immune destruction. Multiple studies have employed spheroids to determine how cancer cells interact with monocytes to create a pro-tumor growth and immunosuppressive tumor microenvironment. Using either PBMCs or cell line-derived monocytes, Raghavan et al. showed that ovarian cancer spheroids cocultured with monocytes were more invasive and less sensitive to chemotherapy in vitro [46]. When these spheroids were inoculated in immunodeficient mice the cocultured spheroids had faster tumor initiation and were less responsive to an IL-6 inhibitor than the homogenous spheroids which lacked monocytes. Using a combination of small molecule inhibitors and shRNA knockdown experiments, Raghaven et al. demonstrated cancer stem cells induce monocyte polarization to the more pro-tumor (M2) phenotype via IL-10 and Wnt signaling. In return, the M2 monocytes promoted cancer stem cell maintenance via IL-6 signaling. Wnt knockdown in M2 monocytes abrogated the increased tumor growth and IL-6 inhibitor resistance—suggesting targeting Wnt signaling may suppress cancer cell induced immunosuppression in melanoma. Monocyte M2 polarization by cancer cells was also seen in pancreatic cancer spheroid models. Kuen et al. showed that spheroids composed of pancreatic cancer cell lines and fibroblasts promoted M2-like polarization of monocytes [47]. Kuen et al. added T cells to the monocyte-fibroblast-cancer cell spheroid cocultures and found that M2 monocytes inhibited CD4+ and CD8+ T cell proliferation and activation. Chandrakesen et al. expanded on the mechanism behind these observations and showed that overexpression of a stem cell marker DCLK1-isoform2 by murine pancreatic cancer cell lines induced M2 polarization via cytokine release [48]. These M2 macrophages go on to inhibit CD8+ T cell proliferation and granzyme B expression. siRNA knockdown of DCLK-isoform2 resulted in greater CD8+ T cell activation and reduced pancreatic cancer cell viability—suggesting DCLK-isoform2 is a novel therapeutic target in pancreatic cancer. These studies highlight how spheroids can be used to elucidate complex cellular crosstalk. Due to the simplicity and controllability of the spheroid coculture systems, these groups could elucidate the detailed mechanisms regulating this complex cellular crosstalk.

Cytokines are well-studied mediators of cancer cell-immune cell crosstalk, but there are additional methods for cellular communication, one of which is release and uptake of extracellular vesicles [49]. Cancer cell-released extracellular vesicles, particularly exosomes, influence many cell types present in the tumor microenvironment including fibroblasts, endothelial cells and stem cells which ultimately promote tumor growth through impacts on cell proliferation, angiogenesis, metabolism, metastasis, and more [50,51]. Cancer cell-released exosomes also impact tumor-resident and distant immune cells and play a critical role in immunotherapy resistance [52,53,54,55,56,57]. Recent studies have highlighted that 3D culture methods may be superior to 2D culture methods in their ability to produce and replicate patient extracellular vesicles. For instance, Rocha et al. reported that in comparison to cells grown in 2D, cells grown in 3D release significantly more extracellular vesicles and the content of the vesicles derived from 3D cultures have increased mircoRNA and decreased protein content [58]. Additionally, the RNA content of extracellular vesicles derived from spheroids are more similar to patient-derived extracellular vesicles than extracellular vesicles derived from 2D cultures [59]. To determine how T and B cells interacted with cancer-derived exosomes, Sadovska et al. used a 3D heterotypic spheroid model that cocultured prostate cancer cell lines that produced GFP+ extracellular vesicles with PBMC-derived T and B cells. They found that B cells interacted with cancer-derived extracellular vesicles more readily than T cells, with approximately 60% of CD19+ B cells testing GFP+ after coculture versus 20% of CD3+ T cells. Furthermore, they suggest that B cells primarily interact with extracellular vesicles on the cell surface whereas a fraction of T cells can internalize extracellular vesicles. These conclusions are different from those of Muller et al., who showed that B cells can internalize cancer-derived extracellular vesicles [60]. The reason for this discrepancy is unclear, but one potential explanation is that Sadovska generated their exosomes using 3D spheroids, while Muller et al. generated their exosomes using 2D cell cultures.

## 3. Organoids

### 3.1. Mechanisms of Immunoediting: Antigen Presentation

Deficient antigen presentation is another mechanism by which cancer cells evade immune destruction [42]. Antigen presentation requires a complex series of cellular interactions dependent on autologous cellular crosstalk. Unlike spheroids, organoids are usually derived from donors (Figure 2) allowing researchers to coculture cancer cells with autologous immune cells. Chakrabarti et al. created a coculture system that consisted of murine derived gastric cancer organoids, autologous spleen/bone marrow derived dendritic cells, and CD8+ T cells [61]. Using this system, they demonstrated that dendritic cells pulsed with organoid conditioned media could activate CD8+ T cells which could kill the organoids. However, dendritic cells that were not pulsed with organoid-conditioned media did not induce CD8+ T cell activation and cytotoxicity. This demonstrates that tumor derived organoids can recapitulate three critical components of antigen presentation: (1) tumor organoids produce tumor antigens, (2) dendritic cells can process and present tumor antigens in organoid models, and (3) CD8+ T cells can be activated by dendritic cells resulting in tumor cell lysis in a TCR-dependent manner. While this work was done using murine cancer and immune cells, Dijkstra et al. used patient-derived colorectal cancer and non-small cell lung cancer organoids to expand autologous circulating T cells [62]. They confirmed the expanded CD8+ T cells were tumor antigen restricted by showing coculture of autologous CD8+ T cells with healthy donor organoids did not significantly activate or expand CD8+ T cells. These studies demonstrate that tumor organoids can be useful models for exploring human antigen presentation. In addition, these models can also be used for expansion of tumor-reactive T cells either for adoptive T cell therapy or T cell engineering.

### 3.2. Fibroblast-Cancer Cell Interactions

The majority of onco-immunology research to date studies immune cell and cancer cell interactions. However, in patient tumors, the stroma is a critical tumor component, consisting of dense extracellular matrix and heterogeneous cell populations that are predominantly fibroblasts. This stroma can comprise as much as 90% of the tumor volume [63]. In some cancer types, patients with more tumor stroma have worse clinical outcomes [64]. While murine models of disease recapitulate fibroblast/extracellular matrix biology, they are limited by: (1) the intrinsic difference between mouse and human, and (2) complex, heterogenous cell populations that restrict assessment of specific fibroblast-cancer cell interactions [65]. Fibroblasts act differently when cultured in 3D versus 2D [66]. For example, when pancreatic cancer derived fibroblasts were cocultured with patient derived pancreatic cancer organoids they maintained alpha smooth muscle actin, a marker of cancer associated fibroblasts in patient tumors, yet alpha smooth muscle actin expression was lost when the fibroblasts were cultured in 2D [65]. Thus, 3D fibroblast cultures more accurately reflect cancer-associated fibroblast biology than 2D cultures. In 3D cultures, fibroblasts enhance organoid growth, invasion, and therapy resistance via both direct cell-cell contact as well as via paracrine signaling [47,67,68,69].

### 3.3. CAR Cell Development

Chimeric antigen receptor (CAR)-modified T cells and NK cells represent a recently emergent pillar of cancer immunotherapy. CARs are engineered receptors that target T or NK cells to recognize and lyse cells that express a specific antigen. The most well-known example is anti-CD19 CAR T cells used to treat acute lymphoblastic lymphoma [70]. CAR cell therapy is successful in liquid malignancies (leukemias), but efficacy in solid malignancies remains limited due to poor target antigen availability, limited CAR cell infiltration into tumors, and tumor microenvironment induced immunosuppression [71]. 2D culture systems can be used to identify novel antigens but are not suitable for accessing tumor infiltration or microenvironment induced immunosuppression. In comparison to 2D cultures, spheroids can also be used to identify novel antigens and assess tumor infiltration but are limited in their ability to recapitulate the complex microenvironments because they are composed of a homogeneous cell type. For instance, Leuci et al. identified CSPG4 as a novel tumor antigen in soft tissue sarcoma [72]. Anti-CSPG4-CAR cells lysed soft tissue sarcomas cell lines in 2D and 3D spheroid models. Using murine xenograft models Leuci et al. confirmed the potential utility of anti-CSPG4-CAR cells in vivo. However, xenograft models require immunodeficient mice and thus do not accurately reflect the clinical environment in which CAR cells are used, because many of the immunosuppressive cell types are absent from the microenvironment. One study by Dillard et al. showed that colorectal cancer cell line Caco-2 form cyst-like structures when grown in 3D matrix similar to organoids [18]. They then demonstrated CD19 CAR T cells could identify and lyse CD19 expressing Caco-2 cysts. By using a cell line instead of patient tissue, Dillard et al. argue this approach could be utilized more broadly in research labs that do not have access to patient tissue. However, this system still does not recapitulate the complex tumor microenvironment that greatly influences CAR cell function.

Unlike spheroids, organoids derived from tumors can maintain the complex multicellular, 3D microenvironment and fill a gap in methodology required for the development of CAR cells against solid malignancies. In a proof-of-concept study, Jacob et al. demonstrated patient derived glioblastoma organoids retained similar histology, microvasculature and cellular heterogeneity as the primary tumors they were derived from [73]. Some of these organoids overexpressed EGFRvIII, and anti-EGFRvIII-CAR T cells successfully infiltrated and killed EGFRvIII overexpressing cancer cells yet incompletely cleared the organoids in culture. In glioblastoma patients treated with anti-EGFRvIII CAR-T cells, CAR-T cells infiltrated the tumor and reduced EGFRvIII expression, suggesting specific lysis of EGFRvIII tumor cells, yet failed to control tumor growth [74]. Therefore, the 3D organoid model appears to better recapitulate the clinical outcomes of anti-EGFR-vIII-CAR T cell therapy in glioblastoma and could potentially be used to interrogate the mechanisms driving incomplete tumor clearance.

Since organoids more accurately reflect clinical response to CAR therapy compared to spheroids or 2D cultures and are more amenable to screening approaches than murine models, Schnalgzer et al. developed a method to use organoids for CAR cell efficacy screening [75]. In this proof-of-concept study, Schnalgzer et al. engineered normal colon and colorectal cancer organoids to express signals detectable by microcopy or spectroscopy such as luciferin or GFP. The addition of cytotoxic CAR cells would induce cell death, ideally exclusively in the cancer-derived organoids, reducing the luminescent or fluorescent signal allowing for dynamic detection of CAR activity over time. They used this model to assess the efficacy of anti-EGFRvIII and anti-Frizzled CAR NK cells. While these methods are preliminary, they represent a promising new approach for CAR cell development and optimization for treatment of solid malignancies.

### 3.4. Personalized Immunotherapy Testing

While immunotherapy provides long lasting remissions for some patients, the majority of patients, approximately 87%, do not respond to treatment [76]. These treatment-unresponsive patients often experience adverse side effects and no clinical benefit, necessitating the development of predictive biomarkers. Predictive biomarkers are certain attributes of the patient or tumor that indicate the patient is more likely to benefit from the therapy than a similar patient who lacks this attribute. Examples of immunotherapy-related predictive biomarkers include tumor mutational burden, PD-L1 expression, and even microbiome characteristics [77]. A second potential approach to improving the prediction of benefit is to test tumor cell sensitivities to treatments ex vivo. Organoids have been used to screen other drug classes, such as cytotoxic drugs and small molecule inhibitors [15,78]. However, testing immunotherapeutic agents in this manner has an additional layer of complexity because the patient’s immune cells must also be factored into the therapeutic screen.

In the previously described tumor derived organoid models, host-derived immune cells in these cultures are either lost rapidly or maintained for about a week before they are rapidly lost prohibiting the application of these methods to assess the efficacy of immunotherapies in these tumors [15,73]. To circumvent this challenge, Votanopoulous et al. created what they termed “immune enhanced patient organoids” that contained melanoma tumor biopsies and paired lymph node specimens co-embedded to form autologous immune cell competent tumor organoids [79]. In six out of seven patient specimens, the organoids correctly reflected clinical response to immune checkpoint inhibitors (anti-PD-1 and anti-CTLA-4).

The Votanopoulous study demonstrates that immune cells collected from locations other than the tumor microenvironment can accurately reflect tumor immune responses in 3D cultures. However, the ideal culture system would utilize tumor infiltrating immune cells as opposed to exogenously added immune cells. In 2018, Neal et al. showed the addition of IL-2 to air-liquid interface organoid culture preserves tumor epithelial cells, stromal components and CD4+ and CD8+ T cells for up to 28 days [80]. Tumor infiltrating T cells in this ex vivo model maintained the same TCR repertoire as the original tumor. Anti-PD-1 antibody induced CD8+ T cell expansion in 83% of organoids derived from anti-PD-1 responsive patients versus 14% of organoids derived from anti-PD-1 nonresponsive patients. While CD8+ T cell expansion is a surrogate measurement of tumor cytotoxicity, these findings suggest these 3D methods can be adapted for more extensive immunotherapy screening.

## 4. Microfluidic Chips

In 2018, Jenkins et al. demonstrated tumor pieces derived from both murine and human tumors embedded in collagen containing 3D microfluidic chips—termed “organotypic tumor spheroids”—continue to grow and maintain immune cell composition [81]. This technique only required 6 days to create final cell clusters, unlike the air-liquid interface method that required 1–2 weeks [80]. A one-week discrepancy may not seem significant, but therapy screening turnaround time is critical for these techniques to be clinically translatable. The authors did not disclose how long they could maintain their organotypic tumor spheroids while also preserving immune and stromal components. Even though these methodologies are relatively recent, subsequent papers have utilized microfluidic chips to recapitulate and interrogate various aspects of tumor immunology.

### 4.1. Mechanisms of Immunoediting: Immune Checkpoints

Immune checkpoints are inhibitory receptors expressed by activated immune cells to facilitate self-tolerance. However, some of these pathways are co-opted by malignant cells facilitating immune evasion. Immune checkpoint inhibitors block these inhibitory receptor-ligand interactions restoring anti-cancer immunity [82]. Several immune checkpoints and checkpoint inhibitors exist, but the most studied and prescribed immune checkpoint inhibitors target PD-1/PD-L1 [83]. Canonically, PD-1 is expressed by T cells and PD-L1 is expressed by tumor cells. However, a subset of PD-L1+ tumoral T cells exists, and their function in cancer biology was unclear until Diskin et al. showed that PD-L1 ligation on PD-L1+ T cells resulted in T cell suppressive signaling [84]. They confirmed that these phenotypic changes were likely clinically relevant after exposure of patient derived organotypic spheroids to PD-L1 ligating Fc resulted in accelerated tumor growth and suppressed intratumoral T cells. This experiment was only feasible because the microfluidic chip-based technique for growing organotypic spheroids maintains tumor infiltrating T cells, including PD-L1+ T cells, as well as other immune cell types that are suppressed by PD-L1+ T cell activation. While murine studies conceivably could have also shown similar findings, it would have taken significantly longer to obtain the results.

Novel combinatorial therapies that enhance anti-PD-1 efficacy in otherwise resistant tumors is also an active area of research [85]. Deng et al. used murine organotypic tumor spheroids to demonstrate that CDK4/6 inhibitors enhance anti-PD-1 efficacy. These findings were replicated in murine cancer models, showing that the 3D culture system reflected the in vivo findings [86]. More studies are needed to determine how accurately organotypic spheroids reflect in vivo responses, but if there is strong correlation these systems can be used to supplement murine preclinical models by shortening the length of study and validating findings with patient-derived tissue.

### 4.2. Migration, Extravasation, and Angiogenesis

Perhaps the most exciting potential of microfluidic chips is the ability to model and investigate tumor and immune cells’ relationships with vasculature. Tumor-associated blood and lymphatic vasculature play a critical role in tumor growth and immune evasion [87]. 2D systems are incapable of creating 3D vascular channels and while murine models obviously have vasculature, the ability to monitor and manipulate cellular interactions with vasculature in real time is limited. Heterotypic spheroid models that incorporate cancer cell lines and fibroblasts or endothelial cells have been used to explore the impact of cancer cells on hypoxia and angiogenesis [17,23]. However, these studies have not investigated the relationships between vasculature and immune cells within a tumor microenvironment. Microfluidic chips provide compartmentalization to culture cancer cells, endothelial cells, and immune cells simultaneously, and therefore address this critical gap in available methodologies. For example, Mascolo et al. used a microfluidic chip to assess Vδ2 T cells extravasation following stimulation with zoledronic acid [88]. They used a “double chamber” chip in which one chamber was coated in endothelial cells (representing the vascular compartment) and the other chamber contained tumor cells embedded in an extracellular matrix (representing the extravascular compartment). They could then monitor Vδ2 T cell extravasation, migration towards and eventual lysis of colorectal cancer cells. Aung et al. also used microfluidic chips to study T cell migration. They embedded breast cancer cell lines, either as spheroids or as dispersed individual cells, with monocytes in a bilayer hydrogel construct, and then added endothelial cells to the outer layer to form a vascular encapsulation. Lastly, they added a T cell line to the outside of the bilayer hydrogel and monitored T cell migration through the vascular layer and into the cancer cell/monocyte layer. They found that both monocytes and hypoxia increased T cell migration into tumors, primarily through alteration of chemoattractant cytokines [89]. This type of model is especially useful for studying immunotherapies in solid malignancies since immune cell infiltration into tumors still remains a major hurdle.

Using intricately engineered microfluidic chips, Cui et al. studied how endothelial cells impacted monocytes and vice versa in glioblastoma [90]. They found that glioblastoma cells induce M2 macrophage polarization, which in turn increased endothelial cell expression of integrin αvβ3 and increased endothelial cell proliferation and angiogenesis. They then showed that inhibiting TGF-β and integrin αvβ3 reduced angiogenesis in their 3D models. These complicated studies of the mechanisms controlling angiogenesis are nearly impossible to execute and analyze in 2D or murine models.

## 5. Additional 3D Culture Methods

### 5.1. Scaffolds

Additional 3D culture methods, such as scaffolds, fall outside of the three broad categories presented (spheroids, organoids and microfluidic chips). Scaffolds are 3D matrices that lack cells and are often used to recapitulate extracellular matrix. Several techniques can be used to create scaffolds including: (1) pre-made porous scaffolds, usually made with natural or synthetic biomaterials; (2) cells which secrete extracellular matrix; and (3) decellularized extracellular matrix derived from tissue (Figure 5) [91]. Scaffolds have been used to study many aspects of cancer biology including invasion, metastases, drug delivery, even as therapeutic agents themselves [92,93,94,95,96].

Engineered porous scaffolds have been used to investigate the impact of the extracellular matrix biophysical properties on immune cells and cancer-immune cell interactions. Wong et al. engineered hydrogels with adjustable stiffness and found that soft extracellular matrix alters stromal cell cytokine and chemokine production, enhancing stromal recruitment of monocytes [97]. Alanso-Nocelo et al. found that stiff matrix increased cancer cell line expression of endothelial-to-mesenchymal transition markers and this expression was further increased upon the addition of macrophages to the cell culture [98].

One elegant study that used cell-secreted matrix investigated the differences between aged and young fibroblast-derived extracellular matrix. This group found that aged extracellular matrix enhances melanoma cell motility, while simultaneous reducing T cell motility [99]. The difference between young and aged patient extracellular matrix may contribute to the shorter overall survival older melanoma patient experience compared to young melanoma patients [100].

To study the impact of cancer-derived matrix versus healthy tissue-derived matrix, some researchers use decellularization techniques to generate patient-derived cell-free extracellular matrix [101]. These types of decellularized biological scaffolds are currently used in tissue engineering and regenerative medicine fields because they facilitate wound healing through a variety of processes such as enhancing proliferation, angiogenesis and modulation of the immune system [102,103]. In colorectal cancer, the proteome and secretome of decellularized patient-derived malignant matrix is different from healthy colon matrix [104]. D’Angelo et al. used these decellularization techniques to obtain matrix from healthy colon biopsies, colon cancer biopsies and metastatic colon cancer liver biopsies and healthy liver biopsies. They then re-cellularized the matrix with cells of the human colorectal cancer line HT-29. They found that HT-29 cells grown in cancer-derived matrix had increased proliferation, migration and reduced sensitivity to common chemotherapies. HT-29 cells grown in metastases-derived scaffolds had increased epithelial-to-mesenchymal transition phenotype [105]. With respect to cancer immunology, Pinto et al. demonstrated that, unlike healthy colon matrix, colorectal cancer-derived matrix polarized macrophages to the anti-inflammatory M2 phenotype, enhancing macrophage production of immunosuppressive cytokines [106]. These studies highlight the impact of extracellular matrix on tumor phenotypes and further demonstrates how 3D culture techniques such as decellularization and recellularization can be used to deepen our understanding of this complicated stroma-cancer-immune cell crosstalk.

Scaffolds can also be used to study immunotherapy. In one study, Wolf et al. implanted a biological scaffold (urinary bladder matrix, which is used clinically for wound management) along with B16-F10 melanoma cells in mice. They found that the addition of urinary bladder matrix inhibited melanoma cell growth in a CD4+ T cell and macrophage dependent manner [103]. Zhang et al. engineered synthetic antigen presenting scaffolds that present signals to T cells in a physiological manner to mediate rapid and controlled T-cell expansion, expediting time to cancer-antigen specific T cell reinfusion [107]. Lastly, many synthetic or biologic scaffolds are being explored to enhance immunotherapy drug delivery and retention at the tumor site [108].

### 5.2. Dynamic Cell Culture

Unlike microfluidic chips that allow for laminar flow through a culture system, dynamic cell culture allows for movement of the cells themselves. These dynamic cell culture systems can be used as techniques to create already discussed models. Spinner bioreactors can be used to create spheroids and organoids. They employ a propeller that continuously spins cells in suspension, thereby reducing cellular adhesion and enhancing homogenous distribution of nutrients and oxygen (Figure 6) [109]. These dynamic cell culture systems can also be used to explore the impact of cellular movement and external forces on cell biology. For example, a rotational bioreactor has a self-rotating culture container that can create microgravity (Figure 6). Microgravity research is more common in space cell biology, but the negative impacts of microgravity on immune cell function has been well documented [110]. The addition of vibration to cell culture systems represents a third approach to turning static 3D culture systems into dynamic culture systems (Figure 6) [111,112].

## 6. Limitations

3D culture systems provide a tremendous opportunity to explore cancer immunology, but they are not without limitations. The increased complexity of 3D systems can make inter- and intra-experiment reproducibility difficult [113]. 3D culture systems are more expensive and less widely available than 2D culture systems. Some microscopy techniques may be unable to image 3D cultures due to culture depth and/or lack of culture transparency [114]. With regards to 3D culture systems to explore cancer immunology, it is difficult to maintain primary immune cells in culture for extended periods of time, regardless of 2D or 3D culturing conditions [15,73]. Furthermore, the incredible complexity of the immune system and requirement of multi-step and multi-cellular interactions may limit the applicability of simple heterotypic or multicellular culture methods. Fortunately, as technology continuously develops, the accessibility, versatility, and relevance of 3D models should also simultaneously expand.

## 7. Conclusions

The past decade has witnessed tremendous strides to advance our understanding of the anti-cancer immune system. These advances span a spectrum from basic to clinical research. As the biology revealed becomes increasingly complex, so too must the models used to study that biology. 3D culture systems have addressed these research needs by providing complex yet interpretable platforms. 3D culture systems have thus far been used to discover new insights into immunotherapy distribution, immune cell penetration, cancer-induced immunosuppression, CAR cell development and much more. There remain additional cancer-immunology related fields where 3D culture systems have yet to be applied, including but not limited to alterations in regulatory T cell migration and activity, myeloid-derived-suppressor cell biology, and vaccine development. As 3D technology cost decreases, accessibility increases, and conceptual applications swell, these technologies will undoubtedly continue to gain popularity—perhaps replacing 2D culture methods altogether.

## Figures and Tables

**Figure 1 cancers-13-00056-f001:**
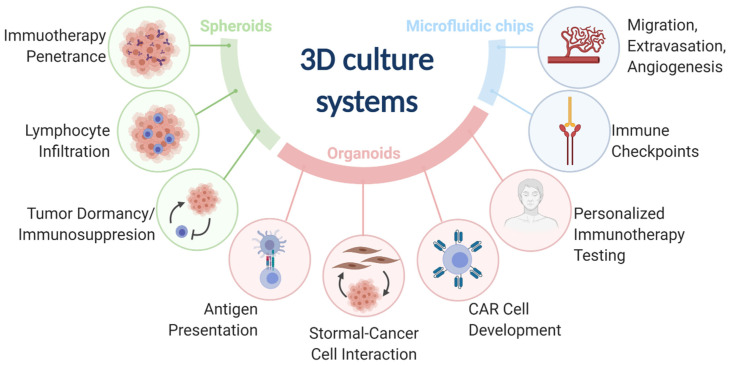
The three main 3D culture systems (spheroids, organoids, and microfluidic chips) and the associated cancer immunology fields that were advanced by the designated 3D culture systems.

**Figure 2 cancers-13-00056-f002:**
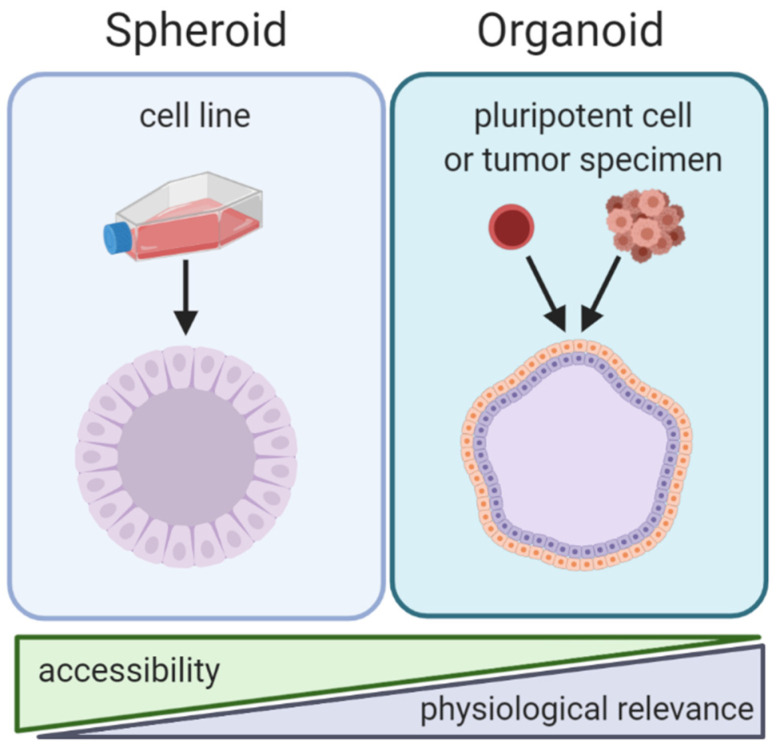
Comparison of spheroid and organoid culture methods. Note: spheroids and organoids in this graphic contain cavities. However, the presence of cavities in spheroids and organoids is variable and some spheroids/organoids are solid.

**Figure 3 cancers-13-00056-f003:**
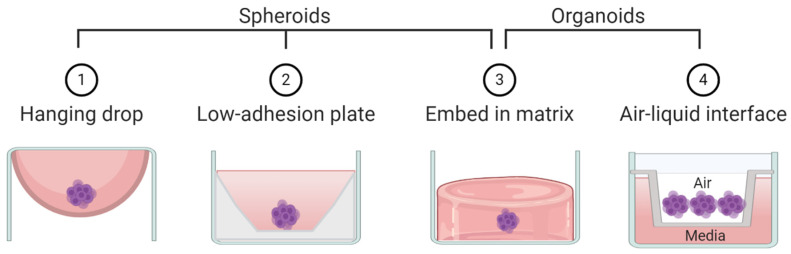
Schematic representation of techniques commonly used to create spheroids and/or organoids.

**Figure 4 cancers-13-00056-f004:**
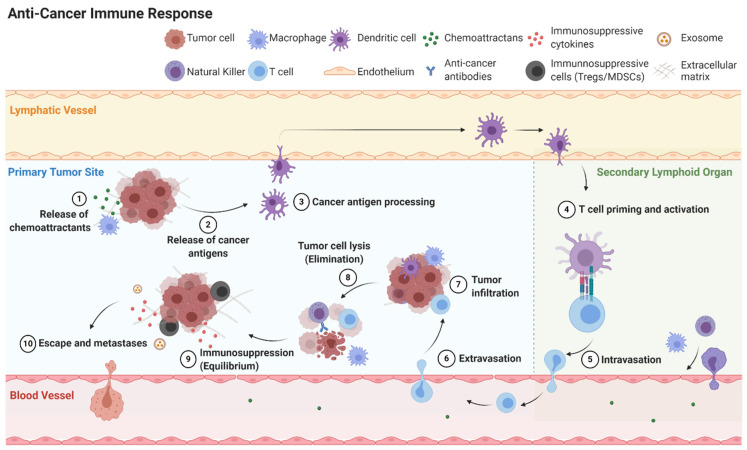
Simplified overview of anti-cancer immune response including elimination, equilibrium and escape.

**Figure 5 cancers-13-00056-f005:**
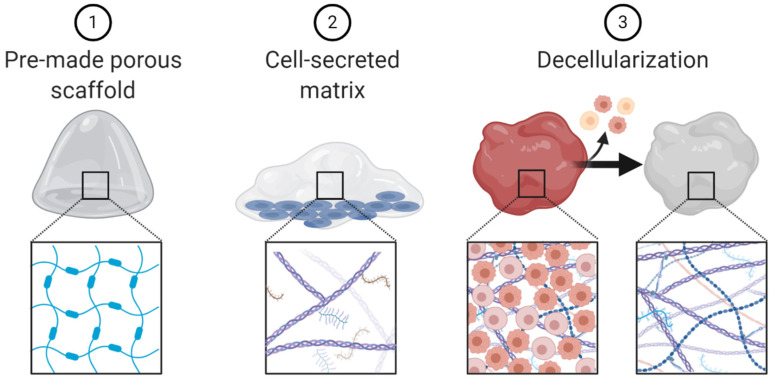
Schematic representation of scaffold engineering techniques.

**Figure 6 cancers-13-00056-f006:**
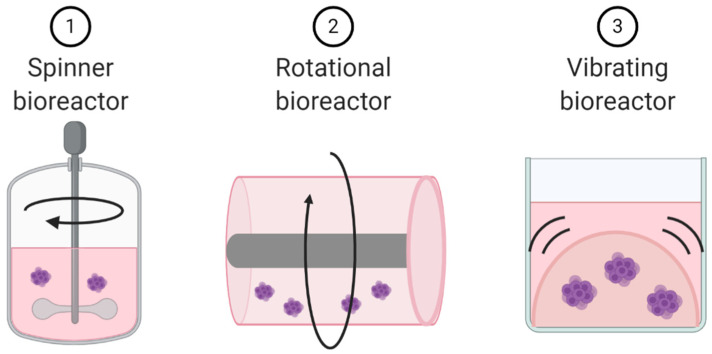
Schematic representation of dynamic cell culture methods.

## Data Availability

No datasets were generated or analyzed in this study.

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
