# Peer review of "3D Culture Systems for Exploring Cancer Immunology"

_cancers, 2020, doi:10.3390/cancers13010056_

Round 1

Reviewer 1 Report

A review of “3D culture systems for exploring cancer immunology” was posted by AA Fitzgerald AA, E Li, and LM Weiner around 2 weeks ago to Cancers (Basel). This review primarily focuses on cancer immunoediting composed of three phases: 1) elimination, 2) equilibrium, and 3) escape. Also, the authors ordered 3D culture systems into 3 categories: i) spheroids, ii) organoids (including tumoroids), and iii) biotechnology (including 3D microfluidic chips, angiogenic/lymphatic vasculatures). The current version was described with 58 references to cover state-of-the-art cancer immunology studies adopting 3D culture systems. However, several factors are lacking in the current version as listed below.

  1. Figures and/or tables must be added to reach more than 5 items. The current 2 figures are simple, conceptual, and understandable. But, only two figures are put on the very end of the Ms. far from the description- must be moved close to the description where they are mentioned. The term “biotechnology” used in Fig 1 is too abstractive and nonobjective. Adding more specific words such as “3D microfluidic chips” will be favorable.
  2. The current version was mainly made from the aspect of immunology but lacking the aspect of cancer cells and cancer stem cells. For example, a recent study reported various types of spheroids/organoids/tumoroids (J Clin Med 2020 9(9)2774; Plos One 2018 13(2)). In addition to the terms “spheroid and organoid”, searching the term “tumoroid” will help adding meaningful references and re-organize the Ms.
  3. In Fig 2, both cartoons of spheroid and organoid are drawn with cavities or acini. However, many studies have shown solid spheroids/tumoroids without cavities (Sci Rep 2016 6, 21174; Front Oncol 2018 8, 376)
  4. “Drug penetrance into solid tumors” was featured in the section of spheroids. Some studies have been done to overcome the drug resistance and penetrance by using 3D culture systems (Mol Cancer Ther 2015 14(6)1504 ; Cancers 2020 12(2)523). Besides, transporter-based drug resistance and/or immune evasion can be added, i.e. ABC-transporters.
  5. Immune evasion is one of the main topics, but recent discoveries in this regard is lacking such as that antibodies and antibody-based drugs can be released from cancer cells or tumors with extracellular vesicles (Mol Cancer 2019 18(1)146; Biology 2020 9(3)47), suggesting a novel mechanism of drug resistance and immune evasion. These studies can be mentioned in the section of immune evasion.
  6. Blood and lymphatic vasculature in 3D cultured systems (organoids and tumoroids) should be described more with more references.
  7. The conclusion is currently poorly described. The current conclusion includes the limitations and advantages of spheroids and organoids (or tumoroids) culture systems, which can be an independent section in the main text. Then, the conclusion must be re-written.
  8. Some words might be miswritten. penetance may be penetrance. ail-liquid may be air-liquid.

Reviewer 2 Report

GENERAL COMMENTS

The subject is topical and the reviewer is aware the topic is challenging to tackle.

The writing style and mistakes sometimes obscure the scientific message. The language used throughout the article is somewhat stilted. There are many careless grammatical and syntactical errors which need to be corrected (some are shown below in the comments).

The balance of explanations of concepts with examples of specific research findings is somewhat uneven throughout all sections.

The manuscript would benefit greatly from another diagram, which illustrates components of the immune system which are pivotal to immunotherapies. This would demonstrate at a glance areas that have been attempted versus those that have not.

Figure 1 would benefit greatly from an accompanying supplementary table to link areas of specific research to the model systems used.

SPECIFIC COMMENTS

SIMPLE SUMMARY

Please correct a number of grammatical errors

ABSTRACT

Please change this sentence: “In this review, we discuss the most common 3D culture  systems—spheroids, organoids, and biotechnology devices—and delve into the novel biology these  culture systems have revealed”.

What do you mean by: delve into the novel biology these culture systems have revealed?

INTRODUCTION:

Line 48: … without her consent

Please add: as was the practice at that time

Lines 52-53: However, this technique still requires growing 52 cells on hard, rigid surfaces …

Please add: made of plastic

Lines 60-61: In these models, malignant cells are injected into the mouse or mice are engineered to develop cancer over a specific 61 course of time or in response to certain stimuli.

Please correct the English

Lines 72-73: While these mice provide a wonderful model to study human immune cell-tumor interactions

Replace the adjective “ wonderful” as it is not the most appropriate in this context

Lines 78-80: 3D models have been used to advance  many fields of cancer immunology research (Figure 1) and can be broadly categorized into scaffold techniques and scaffold-free techniques.  

Models are not techniques. Please correct the English

Lines: 92-93: Some papers describe these tumor-derived multicellular clusters “tumoroids” [8].

The reference used is: Finnberg, N.K.; Gokare, P.; Lev, A.; Grivennikov, S.I.; MacFarlane, A.W.; Campbell, K.S.; Winters, R.M.; 450 Kaputa, K.; Farma, J.M.; Abbas, A.E.-S.; et al. Application of 3D tumoroid systems to define immune and 451 cytotoxic therapeutic responses based on tumoroid and tissue slice culture molecular signatures. Oncotarget 452 2017, 8, 66747–66757, doi:10.18632/oncotarget.19965.

The term tumouroid and specific manufacture has been used extensively by U.Cheema et al – please reference their papers

SPHEROIDS

This section contains a substantial number of tautologies and simple statements which are not needed, such as line 113: Drug development is a core component of contemporary cancer research

The title of 2.2 may be more representative if “lymphocyte penetration” was replied by “immune cell penetration”

Lines 133-134:

In a proof-of-concept study Liu et al. used MALDI-MSI technique to assess cetuximab, an anti-EGFR monoclonal antibody, distribution throughout colon cancer spheroids [18].

It is important that Liu’s findings are stated here.

Line 135: It is reasonable to propose that researchers assess engineered antibodies for enhanced tumor penetration using this method.

This sentence is unnecessary

ORGANOIDS

Line 261: However, in patient tumors a critical tumor component is the stroma, which consists of dense extracellular matrix and fibroblasts

It should be noted that fibroblasts are the main, but not the only cell type in the stroma

Line 278: CAR cell therapy is successful in liquid  malignancies,

Liquid= heamatological?

Line 281: 2D culture systems can be used to identify novel antigens in solid systems, but are not suitable for accessing tumor infiltration or microenvironment induced immunosuppression

What are solid systems?

Line 309: Organoids more accurately reflect clinical response to CAR therapy compared to spheroids or 2D cultures and are more amenable to screening approaches than murine models.

This statement is unnecessary, as the message has been stated before, in various versions

Line 323: Predictive biomarkers are certain attributes of the patient or tumor that indicate the patient is more likely to benefit from the therapy than a similar patient who lacks this attribute

Unnecessary statement

Line 377: These findings were replicated in murine cancer models showing the 3D culture system reflects the in vivo findings

Was this reported in reference 52? If yes, please reference appropriately

FIGURE 1

Figure 1. The three main 3D culture systems (spheroids, organoids, and biotechnology) and the associated cancer immunology fields that were advanced by the designated 3D culture systems.

I suggest a new name for culture system no 3 – not biotechnology. Perhaps biotechnology devices?

Furthermore, the figure could be accompanied by a table (supplementary) where references are tabulated for each culture system and the fields it has supported for clarity and for cross-referencing with the body of the text

FIGURE 2:

Figure 2. Comparison of spheroid and organoid 2D culture methods.

Delete: 2D

REFERENCES

Please correct / update references 59, 60, 61

Reviewer 3 Report

The review article proposed by Fitzgerald A and colleagues addresses an extremely important issue in the panorama of in vitro disease models which they contextualize in the context of immunotherapy and tumor cell-stroma crosstalk.

The article is well planned and written. 

My only concern is the short space given to technologies such as decellularized/recellularized pateint-derived scaffolds as engineered in vitro construct in order to study drug efficency, penetrance and the role of extracellular matrix in the context of cancer onset, growth and diffusion.

Therefore, I would suggest to the authors to improve the "Conclusions" by adding the following citations in order to include such in vitro models:

  • doi: 10.3390/cancers12020364
  • doi: 10.1002/jcp.26403

Finally there are typos at page 3 and 8 to check.

Reviewer 4 Report

The manuscript is well written, sound and comprehensive. The only remark is that no mention has been made of dynamic cell culture models, that in the immunology field could have broad and appropriate application. I think that the addition of a paragraph dealing with dynamic culture methods could be of interest to readers.

Reviewer 5 Report

Fitzgerald and co-authors have composed a review on different 3D culture systems for modelling cancer immunology. The authors discuss the benefits of 3D models as opposed to 2D studies and in vivo models, and the different ways that 3D culture systems have been used to study immune cell interactions in the context of cancer. The manuscript is well written overall, but I have some questions and comments after reviewing the document: Lines 83-85: The current definition of hydrogel types would benefit from some modifications as it is not quite accurate. It would be more appropriate to consider hydrogels as synthetic or natural in origin, and in some cases a mixture of both (semi-synthetic). Natural hydrogels are derived from proteins (such as collagen, gelatin fibronectin) and polysaccharides (such as agarose or alginate). Synthetic hydrogels are chemically synthesised polymers (eg PEG), and semi-synthetic hydrogels arise from some kind of modification to a natural product (e.g. methacrylation of gelatin to form a photo-crosslinkable hydrogel network). Line 85-86: "Organoids are the most common scaffold-based 3D culture method in cancer research". Is there a reference for this claim? It would seem logical that spheroid-based 3F cultures derived from cell lines is more common due to the ease of access and simpler culture method. Following on from this, in line 101-102, i would suggest to change to "Spheroids can be made using low adhesion plates..." As there is a lot of literature also available on the 3D growth of spheroids in various hydrogel systems/ Both spheroids and organoids can be grown in scaffolds and scaffold-free. Perhaps it is better for the authors to describe the differences between spheroids and organoids as they have already nicely done, and also include a sentence or two on the differences between scaffold and scaffold-free culture and how this can be applied in both culture types. Figure 2 would also need to be updated. Overall comments: there are large chunks of text throughout the manuscript that do not have references. The authors should ensure that all statements are supported by referencing the relevant literature. One example: line 262-263 "stroma can comprise as much as 90% of the tumour volume" has no reference.

Round 2

Reviewer 1 Report

The authors corrected all the points I described.

Author Response

Thank you.

Reviewer 2 Report

All comments addressed appropriately and satisfactorily. Thank you for the extra table etc.

One minor mistake -  in the legend of figure 1:

Figure 1. The three main 3D culture systems (spheroids, organoids, and biotechnology) and the 87 associated cancer immunology fields that were advanced by the designated 3D culture 88 systems.

I believe that the text in the brackets should be: spheroids, organoids and microfluidic chips.

Author Response

You are correct! Thank you for catching this error. We have replaced "biotechnology" with "microfluidic chips" in the legend of Figure 1. 

Reviewer 5 Report

The authors have greatly improve the flow and scope of the review article through their extensive revisions.

One additional comment. The text in figure 4 is quite small and difficult to read. Particularly for the cell type "legend" at the top of the figure. Can the authors please modify to figure to increase the text size.

Author Response

Thank you. We have updated figure 4 to increase the text size for all text elements.